# The role of $^{18}$F-FDG PET/CT in patients with diffuse large B-cell lymphoma after radioimmunotherapy using $^{131}$I-rituximab as consolidation therapy

**Joon Ho Choi**[1¤], **Ilhan Lim**[1,2‡*], **Byung Hyun Byun**[1], **Byung Il Kim**[1], **Chang Woon Choi**[1], **Hye Jin Kang**[3‡*], **Dong-Yeop Shin**[4], **Sang Moo Lim**[1]

1 Department of Nuclear Medicine, Korea Cancer Center Hospital, Korea Institute of Radiological and Medical Sciences (KIRAMS), Seoul, Republic of Korea, 2 Department of Radiological & Medico-Oncological Sciences, University of Science and Technology (UST), Seoul, Korea, 3 Division of Hematology and Medical Oncology, Department of Internal Medicine, Korea Institute of Radiological and Medical Sciences (KIRAMS), Seoul, Republic of Korea, 4 Department of Internal Medicine, Seoul National University Hospital, Seoul, Republic of Korea

¤ Current address: Department of Nuclear Medicine, Soonchunhyang University Bucheon Hospital, Bucheon, Republic of Korea
‡ IL and HJK should be considered joint corresponding authors.
* ilhan@kcch.re.kr (IL); hyejin@kirams.re.kr (HJK)

**Data Availability Statement:** All relevant data are within the paper and its Supporting information files.

## Abstract

### Purpose

To evaluate the prognostic value of pretreatment $^{18}$F-FDG PET/CT after consolidation therapy of $^{131}$I-rituximab in patients with diffuse large B-cell lymphoma (DLBCL) who had acquired complete remission after receiving chemotherapy.

### Methods

Patients who were diagnosed with DLBCL via histologic confirmation were retrospectively reviewed. All patients had achieved complete remission after 6 to 8 cycles of R-CHOP (rituximab, cyclophosphamide, vincristine, doxorubicin, and prednisolone) chemotherapy after which they underwent consolidation treatment with $^{131}$I-rituximab. $^{18}$F-FDG PET/CT scans were performed before R-CHOP for initial staging. The largest diameter of tumor, maximum standardized uptake value (SUVmax), metabolic tumor volume (MTV), and total lesion glycolysis (TLG) were obtained from pretreatment $^{18}$F-FDG PET/CT scans. Receiver-operating characteristic curves analysis was introduced for assessing the optimal criteria. Kaplan-Meier curve survival analysis was performed to evaluate both relapse free survival (RFS) and overall survival (OS).

### Results

A total of 15 patients (12 males and 3 females) with a mean age of 56 (range, 30–73) years were enrolled. The median follow-up period of these patients was 73 months (range, 11–108 months). Four (27%) patients relapsed. Of them, three died during follow-up. Median

**Funding:** This project was supported by a grant (no. 50547-2022) from the Korea Institute of Radiological and Medical Sciences (KIRAMS) funded by the Ministry of Science, ICT (MSIT), Republic of Korea. It was also supported by a grant (No. 2020R1A2C2102492) of the National Research Foundation (NRF) funded by the Ministry of Science and ICT (MSIT), Republic of Korea.

**Competing interests:** The authors have declared that no competing interests exist.

values of the largest tumor size, highest SUVmax, MTV, and TLG were 5.3 cm (range, 2.0–16.4 cm), 20.2 (range, 11.1–67.4), 231.51 (range, 15–38.34), and 1277.95 (range, 238.37–10341.04), respectively. Patients with SUVmax less than or equal to 16.9 showed significantly worse RFS than patients with SUVmax greater than 16.9 (5-year RFS rate: 60% vs. 100%, $p$ = 0.008). Patients with SUVmax less than or equal to 16.9 showed significantly worse OS than patients with SUVmax greater than 16.9 (5-year OS rate: 80% vs. 100% $p$ = 0.042).

## Conclusion

Higher SUVmax at pretreatment [18]F-FDG PET/CT was associated with better relapse free survival and overall survival in DLBCL patients after consolidation therapy with [131]I-rituximab. However, because this study has a small number of patients, a phase 3 study with a larger number of patients is needed for clinical application in the future.

## Introduction

Diffuse large B-cell lymphoma (DLBCL) is characterized by an aggressive phenotype. It is the most frequent type of non-Hodgkin lymphoma (NHL), accounting for 30–40% of all NHL cases [1, 2]. Over the last decade, the addition of rituximab to traditional cyclophosphamide, doxorubicin, vincristine, and prednisone (R-CHOP) chemotherapy has led to significant advances and improvements inf treatment response and survival for patients with DLBCL [3, 4]. Despite therapeutic advances with the R-CHOP regimen, about 40% of patients with DLBCL cannot be completely cured with R-CHOP therapy. Owing to the high risk of treatment failure, many researchers have suggested consolidation therapy such as autologous stem cell transplantation (ASCT) [5]. However, many patients experience difficulties upon receiving ASCT due to advanced age or poor performance.

Radioimmunotherapy (RIT) is a treatment option that uses Ytrtrium-90 or Iodine-131 conjugated to anti-CD20 monoclonal antibodies to release high-energy, short distance radiation to tumor cells while avoiding damage to adjacent normal tissues. [90]Y-ibritumomab (Zevalin, Spectrum Pharmaceuticals) is the only RIT currently approved in Europe, and its successful use in consolidation treatment following chemotherapy has been well documented in patients with follicular lymphoma (FL), and chronic lymphocytic leukemia (CLL) [6, 7]. The use of [131]I-tositumomab (Bexxar, GlaxoSmithKline) as a consolidation therapy has also been reported in CLL, advanced DLBCL, and FL [8–10]. Meanwhile, a study using rituximab as a consolidation therapy was conducted in a phase 2 clinical study at our institution [11]. This study is based on the rationale that RIT using [131]I and Zevalin have a similar mechanism for targeting CD20 positive B-Cell lymphoma, but due to different physical and biological properties, a difference in survival rate was expected. In the phase 2 trial study, [131]I-rituximab showed promising efficacy as consolidation treatment for patients with DLBCL.

Using functional imaging such as fluorine-18 2-fluoro-2-deoxy-D-glucose positron emission tomography ([18]F FDG PET) scan, an innovative approach for predicting outcomes of therapy for DLBCL has been developed. In 2007, the revised international Working Group response criteria for malignant lymphoma strongly suggested the use of PET for patients with FDG-avid, curable lymphomas like DLBLCL at 6–8 weeks after completion of therapy for evaluating complete response, for better delineating the extent of disease prior to treatment, and for obtaining the prognostic value of interim PET to predict the overall response to therapy

and long-term outcomes [12, 13]. However, no studies have reported the prognostic ability of pretreatment [18]F-FDG PET/CT using [131]I-rituximab as a consolidation therapy in patients with DLBCL.

Thus, the aim of this study was to evaluate prognostic value of pretreatment [18]F-FDG PET/CT using [131]I-rituximab as a consolidation therapy in DLBCL patients who had acquired complete remission after receiving chemotherapy.

## Materials and methods

### Patient enrollment

We retrospectively reviewed a prospective, single-center phase 2 trial initiated in December 2004 and conducted at the Korea Cancer Center Hospital in the Republic of Korea (WHO-ICTRP ID: KCT0000526) [11]. Protocols of this study were approved by our Institutional Review Board (IRB No.: 2019-02-009), and the requirement to obtain informed consent was waived because this study is a retrospective study in which patients' information is anonymized and this study might provide valuable knowledge to improve the patient care. We enrolled patients who were originally diagnosed with advanced stage (Ann Arbor III or IV) or bulky stage II DLBCL via histologic confirmation.

Inclusion criteria for our study were as follows: adequate performance status (Eastern Cooperative Oncology Group performance status of 0–2), hematology (absolute neutrophil count $\geq$ 1500/μL, hemoglobin level $\geq$ 8.0 g/dL, and platelet count $\geq$ 100,000/μL), and serum chemistries (serum transaminase level less than five times the upper limit of normal, serum total bilirubin level < 4 mg/dL, and a serum creatinine level < 1.5 mg/dL). Our exclusion criteria were as follows: primary central nervous system lymphoma or CNS involvement, recent (< 5 years) history of other malignancies, and hemodynamic instability due to recent (< 12 months) history of severe heart disease. The primary end point was relapse-free survival (RFS) and the secondary endpoint was overall survival (OS).

### [18]F-FDG PET/CT scanning protocols

[18]F-FDG PET/CT scans were carried out before the R-CHOP for initial staging. All patients underwent fasting at least for 6 h before intravenous administration of 7.4 MBq/kg of [18]F-FDG. The specific activity and radiochemical purity of [18]F-FDG were > 37 GBq/μmol and > 98%, respectively. All patients had blood glucose levels checked routinely to ensure that they did not exceed 7.2 mmol/L. Whole-body PET acquisition from the skull base to the thighs started 60 minutes after the [18]F-FDG injection.

### Treatment protocols

Patients who received six or eight cycles of R-CHOP and showed complete response participated in this study. [131]I-rituximab was used for consolidation therapy. It was administered within 8 weeks of the last administration of R-CHOP. Potassium iodide was also provided at least 24 h prior to [131]I-rituximab administration for thyroid protection. Before the administration of unlabeled rituximab, patients received premedications of acetaminophen, diphenhydramine, and a serotonin antagonist. Next, 70 mg of unlabeled rituximab was injected to inhibit peripheral B cells. Then [131]I-rituximab (200 mCi of [131]I conjugated with 30 mg of rituximab) was administered less than 4 h after the administration of unlabeled rituximab. Iodination of rituximab was carried out with Iodo-Beads (Pierce Chemical Co, Rockford, IL, USA) in our radiochemical laboratory, using the method described previously [14]. The patient stayed in the hospital for 5 days. Potassium iodide was provided for 14 days.

## Imaging analysis

Imaging interpretation was based on the consensus of two nuclear medicine physicians. The largest diameter of tumor was examined on axial images of CT for attenuation correction of the PET. Lymphoma involvement in CT images were matched with PET images side by side.

Standardized uptake value (SUV) was defined as the tissue concentration (MBq/mL) of the tracer divided by the amount injected per body weight (MBq/g). MIMvista workstation software version 6.8.3 (MIM Software Inc., Cleveland, OH, USA) was used for measuring the SUVmax, MTV, and TLG of lesions. Patient-based analysis was based on the highest SUVmax when multiple lesions were present. MTV was automatically obtained from the tumor volume combined with FDG uptake with a gradient-based method (PET edge) [15]. TLG was defined as MTV multiplied by SUVmean. The gradient method depends on the starting point at the center of the lesion determined by the operator. When the operator drags out from the lesion center, six axes spread out, providing visual feedback for the stating point. Spatial gradients were assessed along each axis. If a large spatial gradient was discovered, the length of an axis was limited [2, 16, 17].

## Definition of relapse-free survival and overall survival

Relapse was based on imaging results and clinical data. Follow-up period was defined as the period from the date of RIT to the date of the last follow-up or death. Relapse-free survival (RFS) was defined as the time interval from the date of RIT to the date of the first documented recurrence or progression of disease. If there was no documented recurrence, RFS was calculated as the time interval from the date of RIT to the date of last follow-up or death. Overall survival (OS) was defined as the time interval from the date of RIT to the date of the last follow-up or death.

## Statistical analysis

All statistical analyses were performed using the MedCalc software version 16.4.3 (MedCalc Software, Mariakerke, Belgium). Categorical variables are displayed as numbers and percentages. Continuous variables are presented as median values with a range. Five-year survival rates were calculated. We assessed the optimal criteria for the highest sum of sensitivity and specificity in terms of tumor size, SUVmax, MTV, and TLG using receiver operating characteristic (ROC) curve analysis. Kaplan-Meier curve survival analysis was used to assess both RFS and OS. A $P$-value of less than 0.05 was considered statistically significant.

# Results

## Patient characteristics

Clinical information of patient characteristics is presented in Table 1. Three (20%) patients were female. The median age was 56 years (range, 30–73 years). Three (20%) patients were diagnosed with bulky stage II disease. Four and eight patients were diagnosed with stage III and stage IV diseases, respectively. Extranodal involvement was found in eight (53%) patients. Regarding eastern cooperative oncology group (ECOG) performance status, 12 (80%) patients had a score of 1 and three (20%) patients had a score of 2.

## Treatment outcomes and follow-up results

Among the 15 patients who received consolidative [131]I-rituximab, six achieved complete response (CR) after three cycles of R-CHOP and 9 achieved CR after six or eight cycles of R-CHOP. All 15 patients received consolidation RIT after achieving complete response. The

**Table 1. Clinical characteristics of patients.**

| Patient No. | Age (years) | Sex | Stage | extranodal involvement | IPI | Largest tumor size (cm) | Highest SUVmax | MTV | TLG | Location of highest SUVmax |
|---|---|---|---|---|---|---|---|---|---|---|
| 1 | 48 | M | 2 | 0 | 1 | 9.4 | 34 | 264.42 | 6063.31 | left cervical LN |
| 2 | 56 | F | 4 | 2 | 4 | 6.4 | 11.1 | 911.34 | 2829.92 | uterus |
| 3 | 45 | M | 4 | 2 | 2 | 2.2 | 16.9 | 39.82 | 257.83 | right supraclavicular LN |
| 4 | 64 | M | 3 | 0 | 3 | 4.5 | 11.2 | 231.51 | 1277.95 | both cervical LN |
| 5 | 71 | F | 4 | 3 | 5 | 5.3 | 67.4 | 393.65 | 3714.19 | abdominal LN |
| 6 | 44 | M | 2 | 0 | 0 | 7.3 | 17.7 | 55.53 | 650.57 | right cervical LN |
| 7 | 30 | M | 4 | 6 | 3 | 16.4 | 24.8 | 765.55 | 10341.04 | abdominopelvic LN |
| 8 | 73 | F | 4 | 3 | 4 | 3.8 | 24.9 | 282.72 | 3343.07 | left axilla LN |
| 9 | 65 | M | 3 | 0 | 2 | 3.8 | 13.2 | 144.93 | 912.52 | left supraclavicular LN |
| 10 | 50 | M | 4 | 2 | 3 | 3.3 | 17.3 | 965.28 | 8366.97 | left nasal cavity |
| 11 | 51 | M | 3 | 0 | 1 | 2.0 | 20.2 | 78.98 | 772.03 | left cervical LN |
| 12 | 59 | M | 4 | 1 | 1 | 6.3 | 30.9 | 91.28 | 1002.01 | sternum |
| 13 | 67 | M | 2 | 0 | 2 | 7.2 | 13.4 | 169.09 | 907.2 | bilateral cervical LN |
| 14 | 64 | M | 3 | 0 | 2 | 3.0 | 21.3 | 38.34 | 238.37 | right tonsil |
| 15 | 53 | M | 4 | 1 | 1 | 9.7 | 26.1 | 430.65 | 4838.62 | right common iliac LN |

median time from the day of the last R-CHOP administration to RIT was 48 days (range, 32–59 days). The median follow-up period of patients was 73 months (range, 11–108 months). Four (27%) patients relapsed. Of them, three died during follow-up. In one patient, follow-up magnetic resonance imaging showed brain metastasis 2 months after the administration of [131]I-rituximab, and the patient died 9 months later due to disease progression. Another patient had a relapse of the disease 31 months later, and survived for 34 months before he died of disease progression. The last patient survived for 43 months after treatment. A graph of the survival period of all patients is shown in Fig 1.

## Imaging findings

All patients underwent baseline [18]F-FDG PET/CT before receiving R-CHOP treatment. A total of 50 lymph nodes of Ann Arbor classification in 15 patients showed positive FDG findings suggestive of lymphoma. The location was as follows: 10 in cervical and supraclavicular LN; 9 in Waldeyer's ring; 7 in axillary and iliac LN each; 5 in aortic LN; 4 in mediastinal LN; 2 in brachial, mesenteric, inguinal LN and spleen each.

The median largest tumor size was 5.3 cm (range, 2.0–16.4 cm). The distribution of the location of the largest tumor was as follows: supraclavicular LN, 4 patients; axillary LN, 3; neck and mesentery, 2 each; abdomen LN, liver, L-spine and sigmoid colon, 1 each. The median highest SUVmax was 20.2 (range, 11.1–67.4). The distribution of the location of the highest SUVmax was as follows: cervical LN, 5 patients; abdominal LN and supraclavicular LN, 2 each; common iliac LN, axillary LN, uterus, nasal cavity, tonsil and sternum, 1 each. MTV and TLG were also measured with a gradient-based PET segmentation method (PET edge). Median values of MTV and TLG were 231.51 (range, 15–38.34) and 1277.95 (range, 238.37–10341.04), respectively.

## ROC curve analysis

To obtain the cutoff values for survival analysis, ROC analysis was performed using the two endpoints of patient relapse and death. Regarding the detection for relapse of patients, the area

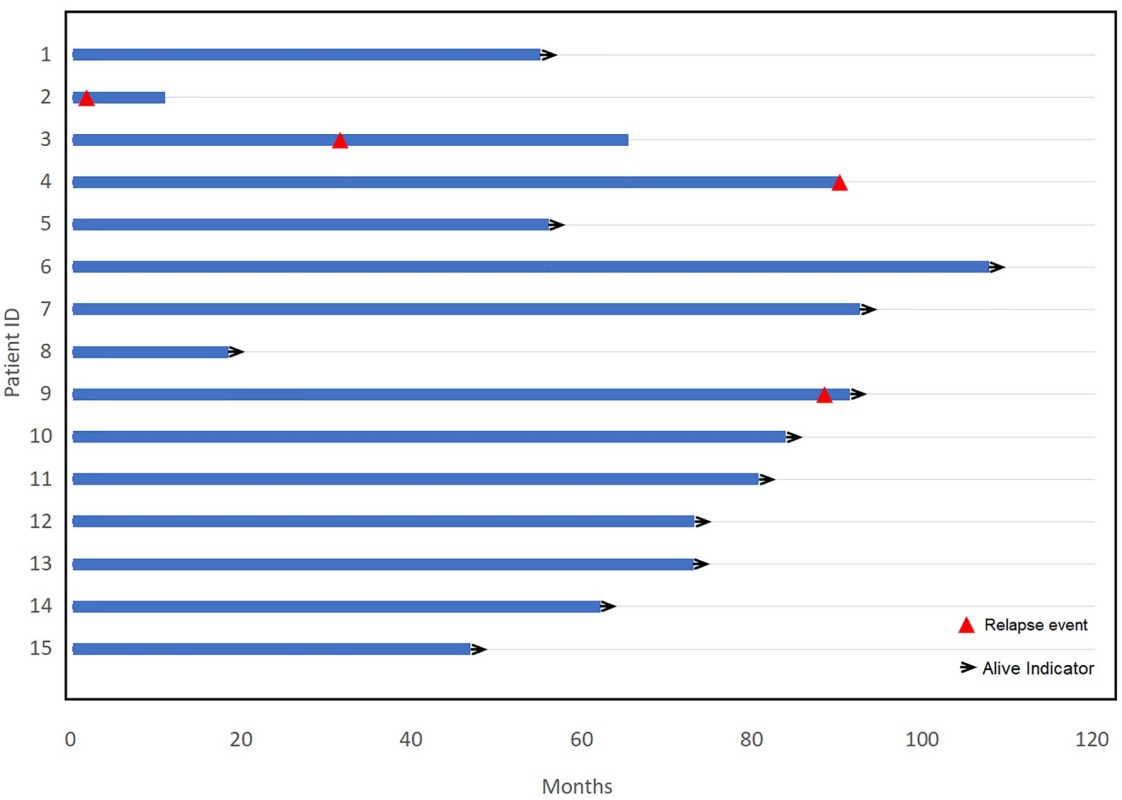

**Fig 1. A swimmer plot showing the survival period of 15 people.** Four people relapsed, of whom three expired. Those who relapsed are indicated by a red triangle, and those who survived are indicated by a black arrow.

under the curve (AUC)s of lesion size, SUVmax, MTV, and TLG were 0.670 (95% CI: 0.387 to 0.884), 0.977 (95% CI: 0.745 to 1.000), 0.545 (95% CI: 0.276 to 0.797), and 0.659 (95% CI: 0.377 to 0.877), respectively. Only SUVmax was found to be statistically significant ($p < 0.001$). The optimal cutoff values for lesion size, SUVmax, MTV, TLG were 6.4, 16.9, 231.51, 2829.92 with sensitivity/specificity of 100%/45%, 100%/91%, 75%/55%, and 100%/55%, respectively.

For discriminating the death of patients, AUCs of lesion size, SUVmax, MTV, and TLG were 0.639 (95% CI: 0.358 to 0.863), 0.944 (95% CI: 0.696 to 0.999), 0.500 (95% CI: 0.239 to 0.761), and 0.639 (95% CI: 0.358 to 0.863), respectively. Only SUVmax was found to be statistically significant ($p < 0.001$). The optimal cutoff values for lesion size, SUVmax, MTV, and TLG were 6.4, 16.9, 39.82, and 2829.92 with sensitivity/specificity of 100%/42%, 100%/83%, 67%/8%, and 100%/50%, respectively.

## Survival analysis

Details of patients' RFS are shown in Table 2 and Fig 2. Median RFS time was 73 months (range, 2–108 months). Two-year and five-year RFS rates were 93% and 86%, respectively. Regarding RFS, only the SUVmax showed a statistically significant difference. Patients with SUVmax less than or equal to 16.9 (n = 5) showed significantly worse RFS than patients with SUVmax greater than 16.9 (n = 10) (Fig 2b; $p = 0.008$, by a log-rank test). Two-year/five-year RFS rates were 80%/60%, for patients with SUVmax less than or equal to 16.9 and 100%/100% for patients with SUVmax greater than 16.9. Tumor size parameter also showed a difference in survival. However, such difference was not statistically significant. Patients with tumor size

**Table 2. Results of univariate Kaplan-Meier survival analysis for relapse free survival.**

| Variable | n | 5-year survival rate | P-value |
|---|---|---|---|
| Median age | | | 0.234 |
| <56 years | 7 (47) | 86% | |
| ≥56 years | 8 (53) | 88% | |
| Sex | | | 0.133 |
| Male | 12 (80) | 92% | |
| Female | 3 (20) | 67% | |
| Ann Arbor stage | | | 0.499 |
| Stage 2 (bulky) | 3 (20) | 100% | |
| Stage 3 | 4 (27) | 100% | |
| Stage 4 | 8 (53) | 73% | |
| IPI | | | 0.176 |
| Low risk (0–1) | 5 (33) | 100% | |
| Intermediate risk (2–3) | 7 (47) | 86% | |
| High risk (4–5) | 3 (20) | 67% | |
| ECOG performance status | | | 0.940 |
| 0 | 0 (0) | not applicable | |
| 1 | 12 (80) | 91% | |
| 2 | 3 (20) | 67% | |
| Largest tumor size (cm) | | | 0.051 |
| ≤6.4 | 10 (67) | 79% | |
| >6.4 | 5 (33) | 100% | |
| Baseline SUVmax | | | 0.008 |
| ≤16.9 | 5 (33) | 60% | |
| >16.9 | 10 (67) | 100% | |
| Baseline MTV (mL) | | | 0.646 |
| ≤231.51 | 8 (53) | 88% | |
| >231.51 | 7 (47) | 86% | |
| Baseline TLG | | | 0.146 |
| ≤2829.92 | 9 (60) | 78% | |
| >2829.92 | 6 (40) | 100% | |

*Numbers in parentheses are percentages.

SUVmax, the maximum standardized uptake value; MTV, metabolic tumor volume; TLG, total lesion glycolysis

less than or equal to 6.4 cm (n = 10) showed worse RFS than patients with tumor size greater than 6.4 cm (n = 5) (Fig 2a; $p = 0.051$ by a log-rank test). Two-year/five-year RFS rates were 75%/50% for patients with tumor size less than or equal to 6.4 cm and 100%/100% for patients with tumor size greater than 6.4 cm.

Details of patients' OS are shown in Table 3 and Fig 3. Median OS time was 73 months (range, 11–108 months). Two-year and five-year OS rates were 93% and 93%, respectively. Female patients (n = 3) showed significantly worse OS than male patients (n = 12) (Table 3; $p = 0.046$ by log-rank test). Two-year/five-year OS rates were 67%/67% for female patients and 100%/100% for male patients. Only the SUVmax showed statistically significant difference. Patients with SUVmax less than or equal to 16.9 (n = 5) showed significantly worse OS than patients with SUVmax greater than 16.9 (n = 10) (Fig 3b; $p = 0.042$, by a log-rank test). Two-year/five-year OS rates were 80%/80% for patients with SUVmax less than or equal to 16.9 and 100%/100% for patients with SUVmax greater than 16.9.

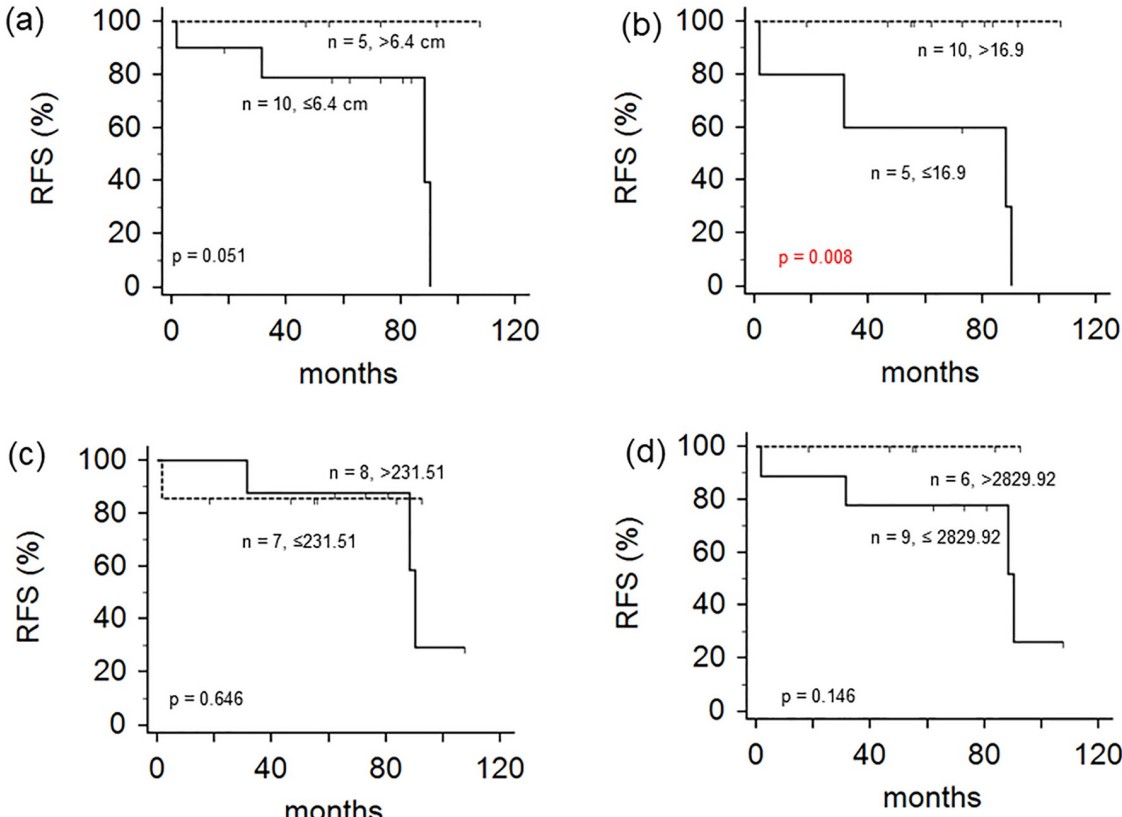

**Fig 2. Kaplan-Meier survival curves depicting relapse-free survival (RFS) according to tumor size (a), the maximum standardized uptake value (SUVmax) (b), metabolic tumor volume (MTV) (c), and total lesion glycolysis (TLG) (d).**

## Case description

Two representative cases demonstrate the relationship between SUVmax from pretreatment $^{18}$F-FDG PET/CT and prognosis. A 45-year-old male was diagnosed with DLBCL. He underwent baseline $^{18}$F-FDG PET/CT before receiving R-CHOP (Fig 4). The largest tumor size was measured to be 2.2 cm in the right supraclavicular LN. The highest SUVmax was 16.9 for the right supraclavicular LN. The patient reached complete remission after six cycles of R-CHOP. Following consolidation RIT, the patient relapsed with secondary hematologic malignancy at 31 months during 65 months of follow-up. He received all-transretinoic acid (ATRA) regimen for additional chemotherapy until he expired.

In contrast, a 71-year-old female underwent baseline $^{18}$F-FDG PET/CT after diagnosis with DLBCL (Fig 5). The largest lymphoma size was 5.3 cm with the highest SUVmax of 67.5 for the abdominal LN. The patient reached complete remission after 8 cycles of R-CHOP. Following consolidation RIT, the patient did not relapse during 56 months of follow-up.

## Discussion

The present study demonstrates that pretreatment $^{18}$F-FDG-PET/CT can be used for predicting survival in DLBCL patients after consolidation therapy with $^{131}$I-rituximab. DLBCL patients with higher SUVmax of baseline $^{18}$F-FDG PET/CT showed better relapse free survival and overall survival after consolidation therapy with $^{131}$I-rituximab. To the best of our knowledge, this is the first study investigating the prognostic value of baseline $^{18}$F-FDG PET/CT in

**Table 3. Results of univariate Kaplan-Meier survival analysis for overall survival.**

| Variable | n | 5-year survival rate | P-value |
|---|---|---|---|
| Median age | | | 0.545 |
| <56 years | 7 (47) | 100% | |
| ≥56 years | 8 (53) | 88% | |
| Sex | | | 0.046 |
| Male | 12 (80) | 100% | |
| Female | 3 (20) | 67% | |
| Ann Arbor stage | | | 0.548 |
| Stage 2 (bulky) | 3 (20) | 100% | |
| Stage 3 | 4 (27) | 100% | |
| Stage 4 | 8 (53) | 88% | |
| IPI | | | 0.102 |
| Low risk (0–1) | 5 (33) | 100% | |
| Intermediate risk (2–3) | 7 (47) | 86% | |
| High risk (4–5) | 3 (20) | 67% | |
| ECOG performance status | | | 0.511 |
| 0 | 0 (0) | not applicable | |
| 1 | 12 (80) | 100% | |
| 2 | 3 (20) | 67% | |
| Largest tumor size (cm) | | | 0.162 |
| ≤6.4 | 10 (67) | 90% | |
| >6.4 | 5 (33) | 100% | |
| Baseline SUVmax | | | 0.042 |
| ≤16.9 | 3 (20) | 80% | |
| >16.9 | 12 (80) | 100% | |
| Baseline MTV | | | 0.103 |
| ≤39.82 | 2 (13) | 100% | |
| >39.82 | 13 (87) | 92% | |
| Baseline TLG | | | 0.26 |
| ≤2829.92 | 9 (60) | 89% | |
| >2829.92 | 6 (40) | 100% | |

*Numbers in parentheses are percentages.

SUVmax, the maximum standardized uptake value; MTV, metabolic tumor volume; TLG, total lesion glycolysis

consolidation therapy using radioimmunotherapy. Results of this study confirmed that we could perform early classification of patients based on the risk of recurrence after consolidation therapy with [131]I-rituximab. This would allow us to optimize the treatment administered to patients. For example, we could improve patient care by applying a more aggressive therapy for high-risk patients.

Previous studies have predicted the prognosis of patients with DLBCL and reported that it is valuable to employ interim PET and the end of treatment PET for predicting the prognosis of patients with DLBCL. Regarding interim PET, many researchers have found that interim PET can play an important role in predicting prognosis [18–23]. However, they applied different parameters, including changes in SUVmax, interim SUVmax [22], tumor to liver ratio [19], and a combination of Deauville Score and International Prognostic Index [20]. With respect to end of treatment PET, many studies have also elucidated that end of treatment PET provides a prognostic value for patients with DLBCL [19, 20, 24]. However, it is not easy to

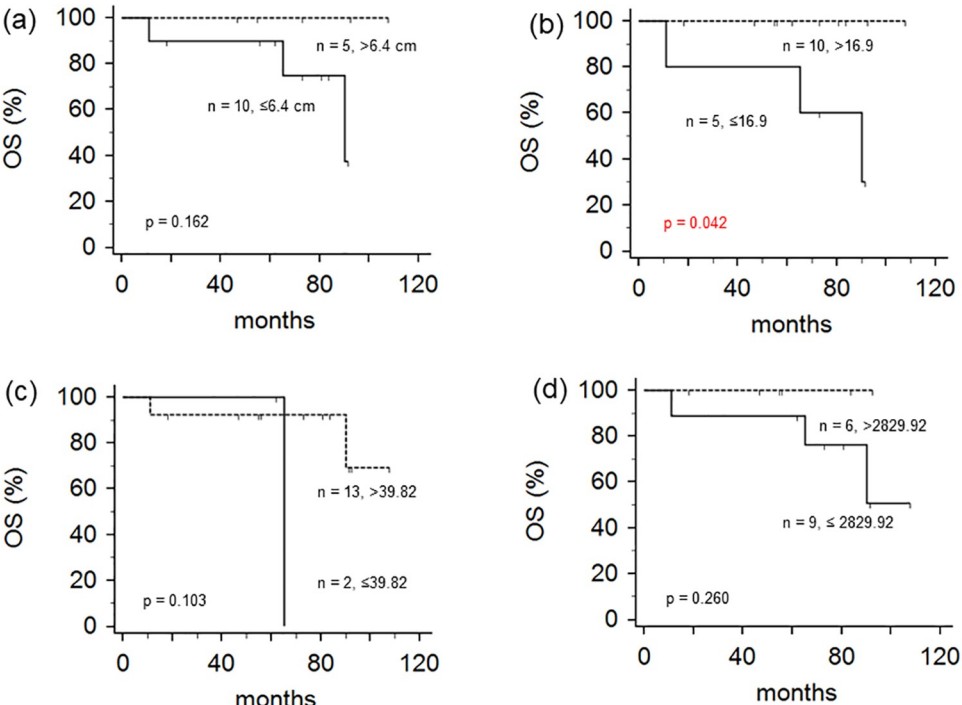

**Fig 3. Kaplan-Meier survival curves depicting overall survival (OS) according to tumor size (a), the maximum standardized uptake value (SUVmax) (b), metabolic tumor volume (MTV) (c), and total lesion glycolysis (TLG) (d).**

apply them for predicting prognosis after consolidation therapy of DLBCL because most patients tend to show complete remission, making it difficult to classify these patients. Therefore, baseline PET might be a better candidate to differentiate the prognosis of patients. Many studies have pointed out that baseline PET shows prognostic implication in patients with DLBCL [22, 25–34]. When utilizing baseline PET, many parameters such as SUVmax, sum of SUVmax, and MTV can be used for analysis.

In our study, it was found that a high SUVmax showed better treatment effect in DLBCL patients who received RIT consolidation therapy. Although, there are not many studies examining the prognostic factors of PET/CT in DLBCL patients who received RIT consolidation therapy after conventional chemotherapy, we compared our results with a study that investigated prognostic factors in patients who had not undergone consolidation therapy. In one meta-analysis, SUVmax and MTV were significant prognostic factors for progression free survival (PFS), but only MTV was reported as a significant factor for OS [35]. Also, in a review study of 20 studies, SUVmax showed significant predictive ability for PFS and OS in 5 studies compared to 17 out of 19 for MTV and 10 out of 13 for TLG showing significant predictive ability for both PFS and OS [36]. A recent study reported that SUVmax was either undetermined or was a prognostic factor only in PFS [37, 38].

To optimize the efficacy of baseline PET, many studies have used metabolic tumor volume [27–34]. These studies showed that higher MTV was associated with worse prognosis of PFS and OS. Similarly, our study revealed that higher MTV tended to be associated with worse prognosis of PFS and OS, although such associations were not statistically significant. In the case of MTV, several issues need to be resolved before MTV can be used as a predictor of prognosis. Until now, there is no standardized methodology for measuring MTV. When it comes

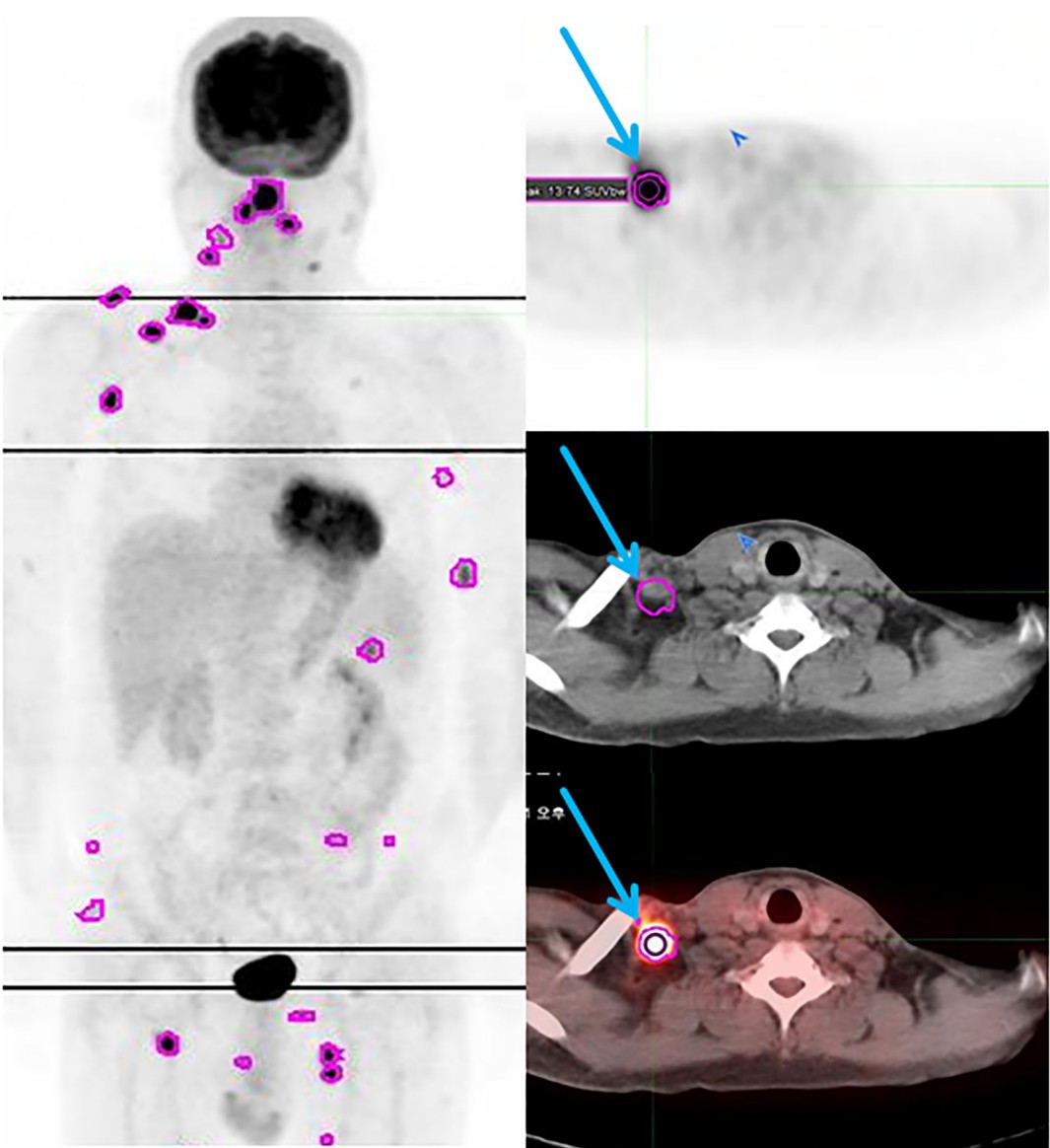

**Fig 4. Lower maximum standardized uptake values (SUVmax) and poor prognosis after radioimmunotherapy (RIT) in a 45-year-old male patient diagnosed with diffuse large B-cell lymphoma (DLBCL).** The patient was at stage 4 with eastern cooperative oncology group (ECOG) performance status of 1, two extranodal involvement, and an intermediate risk of International prognostic index (IPI) score. $^{18}$F-FDG PET/CT showed disseminated hypermetabolic lymph nodes in tonsil, both submandibular, right supraclavicular, both axilla, both inguinal, both arm, and spleen. The largest tumor size was 2.2 cm. The highest SUVmax was 16.9 in the right supraclavicular LN. Metabolic tumor volume (MTV), and total lesion glycolysis (TLG) were 39.82 and 257.83, respectively. The patient experienced recurrence at 31 months after RIT.

to MTV threshold, some researchers employ absolute values such as 2.5 and 3.0, whereas other researchers utilize relative values such as 42%.

Regarding the mechanism of prediction, Gallicchio et al. have demonstrated better survival for patients with DLBCL having higher values of SUVmax who underwent $^{18}$F-FDG PET/CT imaging before receiving chemotherapy [25] ($p$ = 0.0002, HR: 0.13, 95% CI: 0.04–0.46). The present study showed similar results, which was expected because higher level of glycolytic activity was associated with better response to chemotherapy. Furthermore, patients with large

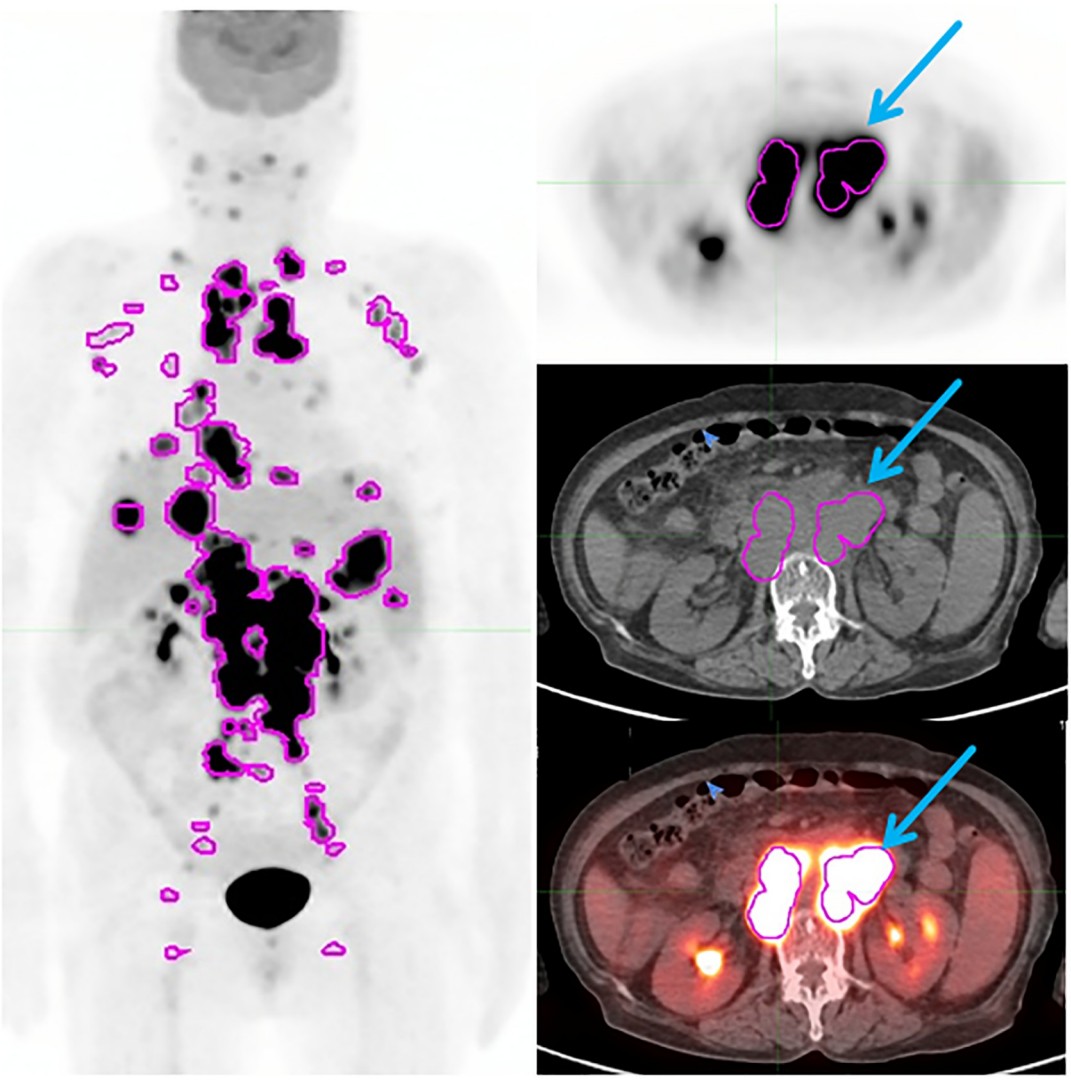

**Fig 5. Higher maximum standardized uptake values (SUVmax) and good prognosis after radioimmunotherapy (RIT) in a 71-year-old female patient diagnosed with diffuse large B-cell lymphoma (DLBCL).** The patient was at stage 4 with eastern cooperative oncology group (ECOG) performance status of 2, three extranodal involvement, and a high risk of international prognostic index (IPI) score. [18]F-FDG PET/CT showed conglomerated hypermetabolic lymph nodes in tonsil, lateral neck, axillary chain, mediastinum, abdominopelvic, liver, and spleen. The largest lymphoma size was 5.3 cm on abdominal LN. The highest SUVmax was 67.4. Metabolic tumor volume (MTV), and total lesion glycolysis (TLG) were 231.51 and 1277.95, respectively. The patient was followed up for 56 months without relapse.

decline of SUVmax in post-treatment PET also have good prognosis in some types of carcinoma [39–41]. This suggests that higher SUV in pretreatment PET may predict better prognosis with good response to treatment. The prognostic value of [18]F-FDG PET/CT in lymphoma has been established [42]. The uptake of [18]F-FDG PET/CT in lymphoma is influenced diversely by multifactorial process of various biological factors such as oncogene expression, apoptosis and viable cell fraction, impacts of hypoxia, extent of inflammatory cell infiltration, and tumor location (nodal/extranodal) [43]. This can lead to uptake heterogeneity in patients with the same histologic subtype [42].

There are some mechanisms that can explain our results. Low SUVs could be associated with hypoxic tumors that can be less aggressive if not treated, but paradoxically more resistant

to chemotherapy treatment [44]. Another possible explanation is that those with higher cell proliferation have higher SUV values [45], and those are more likely to respond to treatment [39, 46]. *In vitro* studies have revealed that cells resistant to chemotherapy show decreased FDG uptake, suggesting that altered glucose transporter may transport into resistant tumor cells [47, 48].

A few studies have examined the prognosis using $^{131}$I-rituximab as RIT with $^{18}$F-FDG PET/CT. Similar to our study, Lim et al. have investigated survival factors using $^{18}$F-FDG PET/CT for the treatment of lymphoma patients with $^{131}$I-rituximab [49]. Their results showed that SUVmax and tumor size were significant predictors. High SUVmax in pretreatment scan was related to poorer overall survival and progression-free survival. They also showed that greater tumor size was associated with poorer OS. However, their study included non-Hodgkin lymphoma patients who were refractory to therapy and patients with low grade lymphoma as well as patients with DLBCL, while our study was limited to DLBCL patients who showed complete remission after R-CHOP therapy. Thus, the different outcomes of the two studies could be the result of the differences in study design. Other studies have investigated prognosis using various types of radionuclides with $^{18}$F-FDG PET/CT [39, 50–52]. Among them, Cazaentre et al. used Y-90 ibritumomab tuixetan and Y-90 epratuzumab tetraxetan as RIT. Their results demonstrated that low SUVmax and low TLG values were associated with good response rates in both groups. However, their study also included other types of NHL patients such as those with mantle cell lymphoma and follicular lymphoma.

The present study has the following limitations. First, only small number of patients were recruited due to low enrollment. This was because the study had a difficulty with enrolling patients for consolidation therapy after complete remission had occurred. Second, non-randomization of the study limits the interpretation of its results. Only patients who received RIT consolidation therapy were included and not the ones who only received chemotherapy. Third, only pretreatment images were obtained. Tumor response could not be evaluated because only patients with CR were enrolled. Fourth, this study has limitations as it is a retrospective study, though data were collected prospectively. Despite these limitations, we consider it important to report our findings in the article in view of the prognostic value of $^{18}$F-FDG PET/CT in RIT consolidation therapy for DLBCL patients.

In conclusion, our result showed higher SUVmax at pretreatment $^{18}$F-FDG PET/CT was associated with better relapse free survival and overall survival in DLBCL patients who received consolidation therapy with $^{131}$I-rituximab. However, since the study was conducted with a small number of subjects and was limited to consolidation therapy patients, a phase 3 trial studies with a larger number of patients are needed before clinical application.

## Supporting information

**S1 Data.**
(XLSX)

## Author Contributions

**Conceptualization:** Ilhan Lim.

**Data curation:** Joon Ho Choi.

**Formal analysis:** Joon Ho Choi.

**Methodology:** Joon Ho Choi, Ilhan Lim, Dong-Yeop Shin.

**Supervision:** Ilhan Lim.

**Validation:** Joon Ho Choi.

**Writing – original draft:** Joon Ho Choi.

**Writing – review & editing:** Ilhan Lim, Byung Hyun Byun, Byung Il Kim, Chang Woon Choi, Hye Jin Kang, Dong-Yeop Shin, Sang Moo Lim.

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
