## [Decision Letter · Decision Letter 0]

28 Jun 2022

PONE-D-22-14199The role of 18F-FDG PET/CT in patients with diffuse large B-cell lymphoma after radioimmunotherapy using 131I-rituximab as consolidation therapyPLOS ONE

Dear Dr. Lim,

Thank you for submitting your manuscript to PLOS ONE. After careful consideration, we feel that it has merit but does not fully meet PLOS ONE’s publication criteria as it currently stands. Therefore, we invite you to submit a revised version of the manuscript that addresses the points raised during the review process.

We look forward to receiving your revised manuscript.

Kind regards,

Domenico Albano

Academic Editor

PLOS ONE

Journal Requirements:

2. In ethics statement in the manuscript and in the online submission form, please provide additional information about the patient records/samples used in your retrospective study. Specifically, please ensure that you have discussed whether all data/samples were fully anonymized before you accessed them and/or whether the IRB or ethics committee waived the requirement for informed consent. If patients provided informed written consent to have data/samples from their medical records used in research, please include this information.

Reviewers' comments:

Reviewer's Responses to Questions

**Comments to the Author**

1. Is the manuscript technically sound, and do the data support the conclusions?

Reviewer #1: Yes

Reviewer #2: No

2. Has the statistical analysis been performed appropriately and rigorously? 

Reviewer #1: Yes

Reviewer #2: No

3. Have the authors made all data underlying the findings in their manuscript fully available?

Reviewer #1: Yes

Reviewer #2: No

4. Is the manuscript presented in an intelligible fashion and written in standard English?

Reviewer #1: Yes

Reviewer #2: No

5. Review Comments to the Author

Reviewer #1: The authors have restrospectively analyzed a prosepctive study of the usefulness of baseline PET/CT in DLBCL patients recieving RCHOP Chemotherapy and Bexxar consolidation therapy.

The manuscript is well-written, and results have been analyzed well.

A few minor points to be addressed:

1. page 14, line 124-125: potassium ioidide was also provided at least 24 hr~. I think the authors should add "for thyroid protection", as some readers may not be familiar with I-131 rituximab treatment protocol.

2. I am not familiar with I-131 rituximab being administered as consolidation therapy. Although it is not possible to compare with other patients who achieved complete remission and did not recieve I-131 rituximab, I think the authors should add a short discription on the possible advantages or reported advantages of this treatment protocol.

Reviewer #2: Choi et al present the results of a ?phase I trial of 131I rituximab as a consolidation therapy following standard of care treatment with 6-8 cycles of RCHOP chemotherapy. The authors report the results of 15 patients (12 males, 3 females) treated with 131I rituximab. The authors conclude that higher pretreatment SUV max predicts better relapse free survival and overall survival following consolidation therapy with 131I rituximab.

The paper has major concerns which would need to be addressed.

1. The details of the study are not sufficient. Was this a phase I study? What was the rationale for the study? Consolidation for DLBCL is not standard of care? How were patients selected and why? What were the intended benefits? What ethical approval was obtained and what is the clinical trial number? More details on trials/proposed consolidation therapy need to be included.

2. In the treatment outcomes and follow up section it mentions three patients died during follow up? What did they did of? Progressive disease or another reason? (line 186)

3. Two case descriptions are given (lines 259-291). Whilst interesting this is not a standard way to report outcomes in clinical trial. A swimmers plot including all patients enrolled with details about outcomes (relapse/death etc) would be more appropriate.

4. I do not agree with the conclusions or they are not explained adequately. The authors conclude that PET/CT pretreatment can be used to predict survival in DLBCL in patients receiving consolidation therapy. Does pretreatment PET/CT not predict outcomes for all patients regardless of whether they received consolidation therapy or not? It would be useful to know the impact of SUVmax pretreatment on survival outcomes of those who don't receive consolidation therapy (either from the literature or comparator cohort from the same institution). The authors conclude that the sex of the patient is associated with prognosis yet there are only 3 females in the cohort! These numbers and the skewed sex ratio of the cohort make it impossible to make any associations between sex and outcomes.

5. The limitations section needs to be significantly expanded and rewirtten. The context of the study is not clear and the list of limitations is long.

Overall the standard of English needs to be significantly improved, particularly the discussion section.

6. PLOS authors have the option to publish the peer review history of their article (what does this mean?). If published, this will include your full peer review and any attached files.

Reviewer #1: No

Reviewer #2: No

---

## [Author Response · Author response to Decision Letter 0]

8 Jul 2022

Dear Dr. Chenette:

I thank the editors and referees of the ‘Plos One’ for taking their time to review my article.

I have made some corrections and clarifications in the manuscript after going over the referee’s comments. And I made a point-by-point reply to reviewer’s comments.

I have also checked the all PLOS ONE's style requirements. I have uploaded the minimal data set that fully anonymize all patient record.

I have specified the informed consent that was waived in the materials and methods section.

I hope the revised manuscript will better meet the requirements of the ‘Plos One’ for publication. I thank you again for the constructive review by the referees.

Sincerely,

Ilhan Lim, M.D., Ph.D.

Department of Nuclear Medicine, Korea Cancer Centre Hospital, Korea Institute of Radiological and Medical Sciences (KIRAMS), 75 Nowon-ro, Nowon-gu, Seoul, Korea

Tel: 82-2-970-1273; Fax: 82-2-970-2438; E-mail: ilhan@kcch.re.kr

Hye Jin Kang, M.D., Ph.D.

Division of Hematology and Medical Oncology, Department of Internal Medicine, Korea Institute of Radiological and Medical Sciences (KIRAMS), 75 Nowon-ro, Nowon-gu, Seoul, Korea

Tel: 82-2-970-1289; E-mail: hyejin@kirams.re.kr

---

## [Decision Letter · Decision Letter 1]

19 Jul 2022

PONE-D-22-14199R1The role of 18F-FDG PET/CT in patients with diffuse large B-cell lymphoma after radioimmunotherapy using 131I-rituximab as consolidation therapyPLOS ONE

Dear Dr. Lim,

Thank you for submitting your manuscript to PLOS ONE. After careful consideration, we feel that it has merit but does not fully meet PLOS ONE’s publication criteria as it currently stands. Therefore, we invite you to submit a revised version of the manuscript that addresses the points raised during the review process.

We look forward to receiving your revised manuscript.

Kind regards,

Domenico Albano

Academic Editor

PLOS ONE

Additional Editor Comments :

The are some doubts about ethical conditions of this study, as reviewer 2 written.

Reviewers' comments:

Reviewer's Responses to Questions

**Comments to the Author**

1. If the authors have adequately addressed your comments raised in a previous round of review and you feel that this manuscript is now acceptable for publication, you may indicate that here to bypass the “Comments to the Author” section, enter your conflict of interest statement in the “Confidential to Editor” section, and submit your "Accept" recommendation.

Reviewer #1: All comments have been addressed

Reviewer #2: (No Response)

2. Is the manuscript technically sound, and do the data support the conclusions?

Reviewer #1: Yes

Reviewer #2: Partly

3. Has the statistical analysis been performed appropriately and rigorously? 

Reviewer #1: Yes

Reviewer #2: Yes

4. Have the authors made all data underlying the findings in their manuscript fully available?

Reviewer #1: Yes

Reviewer #2: No

5. Is the manuscript presented in an intelligible fashion and written in standard English?

Reviewer #1: Yes

Reviewer #2: No

6. Review Comments to the Author

Reviewer #1: The authors have sufficently addressed to my questions.

The authors have elaborated on I-131 rituximab for consolidation in the introduction.

Reviewer #2: The authors have made substantial improvement to the manuscript. However I still have some major concerns. These mainly relate to how the study was conducted. In particular the sentence that states 'the requirement for informed consent was waived'. I cannot understand how a prospective trial of an investigational medicinal product could be conducted without informed consent from the participants. This goes against the Declaration of Helsinki and needs to be clarified before this manuscript can be considered for publication. In addition the standard of English is still poor and needs to be improved.

7. PLOS authors have the option to publish the peer review history of their article (what does this mean?). If published, this will include your full peer review and any attached files.

Reviewer #1: No

Reviewer #2: No

---

## [Author Response · Author response to Decision Letter 1]

1 Aug 2022

We would like to resubmit this manuscript [PONE-D-22-14199R1] - [EMID:3c112168cfebaa1e], entitled “The role of 18F-FDG PET/CT in patients with diffuse large B-cell lymphoma after radioimmunotherapy using 131I-rituximab as consolidation therapy” to PLOS ONE for publication. We appreciate the careful reviews provided by the editor and reviewers. We are submitting a revised version of our manuscript in which we have considered or responded to all of the comments and questions. Our detailed responses are given below. We have also replied to the reviewers’ comments in a point-by-point manner, and the changes made have been highlighted in red in the revised manuscript.

---

## [Decision Letter · Decision Letter 2]

8 Aug 2022

The role of 18F-FDG PET/CT in patients with diffuse large B-cell lymphoma after radioimmunotherapy Dear Dr. Lim Thank you for submitting your manuscript to PLOS ONE. After careful consideration, we feel that it has merit but does not fully meet PLOS ONE’s publication criteria as it currently stands. Therefore, we invite you to submit a revised version of the manuscript that addresses the points raised./>==============================

We look forward to receiving your revised manuscript.

Kind regards,

Domenico Albano

Academic Editor

PLOS ONE

Journal Requirements:

Additional Editor Comments:

Only some minor points as suggest by reviewer need to be corrected

Reviewers' comments:

Reviewer's Responses to Questions

**Comments to the Author**

1. If the authors have adequately addressed your comments raised in a previous round of review and you feel that this manuscript is now acceptable for publication, you may indicate that here to bypass the “Comments to the Author” section, enter your conflict of interest statement in the “Confidential to Editor” section, and submit your "Accept" recommendation.

Reviewer #2: (No Response)

2. Is the manuscript technically sound, and do the data support the conclusions?

Reviewer #2: Yes

3. Has the statistical analysis been performed appropriately and rigorously? 

Reviewer #2: Yes

4. Have the authors made all data underlying the findings in their manuscript fully available?

Reviewer #2: Yes

5. Is the manuscript presented in an intelligible fashion and written in standard English?

Reviewer #2: Yes

6. Review Comments to the Author

Reviewer #2: The manuscript has been substantially improved and the proofreading has improved the quality of the manuscript. I appreciate that consent may have been waived for the retrospective analysis in this manuscript but the authors still need to detail the consent process, ethical approvals for use of data in this way in order to make the manuscript suitable for publication.

7. PLOS authors have the option to publish the peer review history of their article (what does this mean?). If published, this will include your full peer review and any attached files.

Reviewer #2: No

---

## [Author Response · Author response to Decision Letter 2]

9 Aug 2022

We would like to resubmit this manuscript [PONE-D-22-14199R2] - [EMID:bb003148bf32d220], entitled “The role of 18F-FDG PET/CT in patients with diffuse large B-cell lymphoma after radioimmunotherapy using 131I-rituximab as consolidation therapy” to PLOS ONE for publication. We appreciate the careful reviews provided by the editor and reviewers. We are submitting a revised version of our manuscript in which we have considered or responded to all of the comments and questions. Our detailed responses are given below. We have also replied to the reviewers’ comments in a point-by-point manner, and the changes made have been highlighted in red in the revised manuscript.

---

## [Editor Report · Decision Letter 3]

17 Aug 2022

The role of 18F-FDG PET/CT in patients with diffuse large B-cell lymphoma after radioimmunotherapy using 131I-rituximab as consolidation therapy

PONE-D-22-14199R3

Dear Dr. Lim,

We’re pleased to inform you that your manuscript has been judged scientifically suitable for publication and will be formally accepted for publication once it meets all outstanding technical requirements.

Kind regards,

Domenico Albano

Academic Editor

PLOS ONE
---

## [Editor Report · Acceptance letter]

16 Sep 2022

PONE-D-22-14199R3 

The role of 18F-FDG PET/CT in patients with diffuse large B-cell lymphoma after radioimmunotherapy using 131I-rituximab as consolidation therapy 

Dear Dr. Lim:

I'm pleased to inform you that your manuscript has been deemed suitable for publication in PLOS ONE. Congratulations! Your manuscript is now with our production department. 

Kind regards, 

on behalf of

Dr. Domenico Albano 

Academic Editor

PLOS ONE